

# Microbial communities of *Schisandra sphenanthera* Rehd. et Wils. and the correlations between microbial community and the active secondary metabolites

Xiaolu Qin, Han Pu, Xilin Fang, Qianqian Shang, Jianhua Li, Qiaozhu Zhao, Xiaorui Wang and Wei Gu

National Engineering Laboratory for Resource Development of Endangered Crude Drugs in Northwest China, The Key Laboratory of Medicinal Resources and Natural Pharmaceutical Chemistry, The Ministry of Education, College of Life Sciences, Shaanxi Normal University, Xi'an, Shaanxi, China

## ABSTRACT

**Background**. *Schisandra sphenanthera* Rehd. et Wils. is a plant used in traditional Chinese medicine (TCM). However, great differences exist in the content of active secondary metabolites in various parts of *S. sphenanthera*. Do microorganisms critically influence the accumulation of active components in different parts of *S. sphenanthera*?
**Methods**. In this study, 16S/ITS amplicon sequencing analysis was applied to unravel microbial communities in rhizospheric soil and different parts of wild *S. sphenanthera*. At the same time, the active secondary metabolites in different parts were detected, and the correlation between the secondary metabolites and microorganisms was analyzed.
**Results**. The major components identified in the essential oils were sesquiterpene and oxygenated sesquiterpenes. The contents of essential oil components in fruit were much higher than that in stem and leaf, and the dominant essential oil components were different in these parts. The dominant components of the three parts were $\gamma$-muurolene, $\delta$-cadinol, and trans farnesol (stem); $\alpha$-cadinol and neoisolongifolene-8-ol (leaf); isosapathulenol, $\alpha$-santalol, cedrenol, and longiverbenone (fruit). The microbial amplicon sequences were taxonomically grouped into eight (bacteria) and seven (fungi) different phyla. Community diversity and composition analyses showed that different parts of *S. sphenanthera* had similar and unique microbial communities, and functional prediction analysis showed that the main functions of microorganisms were related to metabolism. Moreover, the accumulation of secondary metabolites in *S. sphenanthera* was closely related to the microbial community composition, especially bacteria. In endophytic bacteria, *Staphylococcus* and *Hypomicrobium* had negative effects on five secondary metabolites, among which $\gamma$-muurolene and trans farnesol were the dominant components in the stem. That is, the dominant components in stems were greatly affected by microorganisms. Our results provided a new opportunity to further understand the effects of microorganisms on the active secondary metabolites and provided a basis for further research on the sustainable utilization of *S. sphenanthera*.

Corresponding authors
Xiaorui Wang,
xiaoruiwang@snnu.edu.cn
Wei Gu, weigu@snnu.edu.cn

## INTRODUCTION

The relationship between microorganisms and plants or animals has been one of the most studied research areas in biology or microbiology in recent years, including studies of human gut microfauna (*Liu et al., 2021*) and plant microbiome (*Bai et al., 2015*). In most cases, microorganisms maintain a close mutualistic relationship with animals and plants, through which they obtain nutrients, and the related microbial community can play an important role in the immune system of animals and plants (*Hacquard et al., 2015*). Plant-microbe interactions are important to better understand their role in plant growth and development (*Verma, Chen & White, 2022*). Within plant microbiome research, most attention has been dedicated to endophyte and rhizospheric soil microorganisms (*Bai et al., 2015*; *Zhu et al., 2020*). Detection of rhizospheric soil microorganisms and endophytes in plants has been reported using high-throughput sequencing methods (*Hou et al., 2022*; *Sun et al., 2022*).

The active secondary metabolites in most medicinal plants are affected by environmental and genetic. Endophytes are an important part of the internal environment of medicinal plants, and rhizospheric soil microorganisms are a significant part of the external environment of plants (*Korenblum, Massalha & Aharoni, 2022*; *Hou et al., 2022*). Microorganisms that colonize plant organs and do not cause obvious plant diseases are endophytes (*Shao et al., 2023*). The diversity and community structures of endophytes are closely related to the species, growth stage, different parts, living environment, and genotype of the host plants (*Xia et al., 2023*). In addition, endophytes can produce antibiotics, enzymes, plant growth regulators, alkaloids, and a series of metabolites (*Shao et al., 2023*). The rhizosphere environment is an important place for plant growth, metabolism, and absorption of soil nutrients, as well as the most direct interaction between roots and soil (*Li et al., 2023*). Rhizospheric soil contains a large number of rhizosphere microorganisms, whose quantity and diversity are significantly higher than that of bulk soil (*Yuan et al., 2022*). Rhizospheric soil microorganisms can increase plant tolerance to biotic and abiotic stresses, improve soil nutrient absorption, and affect plant yield and quality (*Vries et al., 2020*). Furthermore, there are interactions between rhizospheric soil microorganisms and plant endophytes (*He et al., 2021*). They can increase the active components of medicinal plants by producing products that are the same or similar to the active secondary metabolites of medicinal plants, or by transforming the original active secondary metabolites of medicinal plants into new compounds to increase the types of active components (*Korenblum, Massalha & Aharoni, 2022*; *Hou et al., 2022*). Therefore, the study of endophytes and rhizospheric soil microorganisms of medicinal plants, as well as the influence of microorganisms on active secondary metabolites, has become an important content in the production of Chinese medicine and the development of new drugs.

*Schisandra sphenanthera* Rehd. et Wils., a perennial deciduous woody vine of the genus *Schisandra*, which is a high-value traditional Chinese medicine (TCM) (*Editorial Committee of Flora of China, 1996*; *Smith, 1947*; *The State Pharmacopoeia Commission of P. R. China, 2020*). *S. sphenanthera* grows on moist mountain slopes or shrubs at an altitude

of 600–3,000 m and is distributed in Shanxi, Shaanxi, Gansu, Sichuan, and other regions of China (*Editorial Committee of Flora of China, 1996*; *Smith, 1947*). The dried and ripe fruit is typically used as medicine, known as "Nanwuweizi", to treat chronic cough, asthma, night sweats, and palpitations insomnia (*The State Pharmacopoeia Commission of P. R. China, 2020*). The active ingredients in fruits include lignans, essential oils, polysaccharides, *etc.* (*Gu, Wei & Wang, 2008*; *Lu et al., 2012*; *Wang et al., 2018*). Pharmacological studies have indicated that *S. sphenanthera* possessed hepatoprotective effects, anticancer activity, central neuroprotective effects, antioxidative activity, *etc* (*Yang et al., 2022*). In addition, the fruits of *S. sphenanthera* can also be used in cosmetics and health products, and listed as a functional food by the Ministry of Health of the P. R. China (*Huang et al., 2021*). Since its main source is wild resources, the genetic improvement and extensive commercial use of *S. sphenanthera* are limited by the depletion of natural herbal resources.

At present, the research of *S. sphenanthera* mainly focuses on the separation and extraction of chemical components and pharmacological activities (*Yang et al., 2022*). In terms of the ecological environment, most studies focus on the effects of environmental factors (climate and soil characteristics) on the active components of *S. sphenanthera* and their potential spatial distribution (*Guo et al., 2016*; *Lu et al., 2012*). However, the composition, diversity, and function of the microbial communities of endangered wild *S. sphenanthera*, and their effects on the accumulation of active secondary metabolites, were still unclear. The specific endophytes isolated from *S. sphenanthera* by You et al. can promote the growth of the host (*You et al., 2021*). However, up to now, minimal studies are available that involve the *S. sphenanthera* endophytes, both domestically and abroad.

In the present study, we focused on two main questions: (i) How variable are rhizospheric soil microorganisms and endophytes in different parts of *S. sphenanthera*? (ii) Do the microbial communities influence the accumulation of active secondary metabolites? Accordingly, this study evaluates the diversity and composition of microorganisms in the rhizospheric soil, root, stem, leaf, and fruit of wild *S. sphenanthera* by high-throughput 16s rRNA and ITS amplicon sequencing. Furthermore, the correlation between the active secondary metabolites and the microorganisms of *S. sphenanthera* is also investigated. The purpose of this study is to provide a reference for the exploitation of microorganisms and the improvement of the yield and quality of medicinal plants by analyzing the interaction between the active secondary metabolites and the microorganisms of *S. sphenanthera*.

## MATERIALS & METHODS

### Sampling and sample processing

Portions of this text were previously published as part of a preprint (https://www. researchsquare.com/article/rs-1937757/v1). Healthy plant materials (*S. sphenanthera*) were collected from Zhashui County, Shaanxi province of China, south of the Qinling Mountains, in August 2019. Rhizospheric soil, root, stem, leaf, and fruit of five individual plants lying approximately 10 m apart were sampled using sterile tools. The roots of similar thickness (about two mm) were collected at a depth of 10–20 cm below the ground, without damaging the taproot. The rhizospheric soil was soil particles adhered to the root system.

The residual soil remaining on the root system was collected after gently shaking off the bulk soil attached to the root system (*Jiao et al., 2022*; *Song et al., 2023*). For stem and leaf, one complete branch was randomly collected from each plant, and all leaves were collected from the sampled offshoot. Fruits were collected from each individual. The plant materials were transported to the laboratory with the sterile bag in ice boxes, and stored at −80 °C. All voucher specimens of plant materials were stored in the College of Life Sciences, Shaanxi Normal University (Voucher number: SN-ZS-YP-ZJW 001-003).

Part of the sample was air-dried in the shade at room temperature, while the other part was cleared and sterilized from epiphytic bacteria (surface sterilization) according to the following methods. 5 g of root, stem, leaf, and fruit were divided into small pieces with a sterile scalpel, and soaked in 75% ethanol for 3 min. Then, the surface-sterilized samples were washed 3–5 times with sterile water to eliminate excess ethanol. To evaluate the efficiency of the surface sterilization, 100 µl of the past washed distilled water was plated on trypsin soybean agar (TSA) and potato dextrose agar (PDA) plates, and plates were incubated at 28 °C and 35 °C, respectively. Finally, samples corresponding to plates without bacterial and fungal growth were used for sequencing.

## DNA extraction, PCR amplification, high-throughput sequencing, and sequencing data processing

The DNA of the root, stem, leaf, and fruit was extracted by the cetyltrimethylammonium bromide (CTAB) method. The DNA of rhizospheric soil microorganisms was extracted using Soil Kit (Thermo Fisher Scientific, Waltham, MA, USA) following the manufacturer's instructions. Primers designed according to conserved regions (bacterial 16S rRNA gene primer (V4) and fungal ITS1-5F rRNA gene primer) were used for the community analysis of bacteria and fungi (Table S1). PCR reactions were performed with 10 ng of template DNA, 2 µM of primers, and 15 µL of Phusion High Fidelity PCR Master Mix (New England Biolabs, Ipswich, MA, USA). The cycling parameters of PCR were initiated at 98 °C for 1 min, followed by 30 cycles (98 °C for 10 s, 50 °C for 30 s, 72 °C for 30 s), and finally extended at 72 °C for 5 min. After the detection of PCR products with 2% agarose gel, PCR products were purified, quantified, and homogenized to form the microbial sequencing library.

The purified amplicon libraries were pooled in equal concentration and sequenced with a Miseq (Illumina, Foster City, CA, USA) system following the manipulation instructions at SAGENE Guangzhou (China). Sequences with ≥97% similarity were assigned to the same OTUs and were analyzed by Uparse software (Uparse v7.0.100). The SILVA database was used based on the Mothur method to annotate OTU sequences. To investigate the phylogenetic relationships of different OTUs, multiple sequence alignment was performed using MUSCLE software (Version 3.8.31). Subsequent analyses of alpha diversity and beta diversity were performed after the normalization of OTU abundance information. Alpha diversity (Chao1 index, ACE index, Shannon, and Simpson index) in our samples was calculated with QIIME (v1.9.1), which was used to analyze the complexity of species diversity of rhizospheric soil and four parts (root, stem, leaf, and fruit). Chao1 and ACE were used to evaluate the microbial species richness. Shannon and Simpson were used to

evaluate the community diversity of microflora. Beta diversity analysis was also calculated with QIIME.

## Extraction of essential oils and component identification

As too many root samples would cause damage to plants, the essential oil components were not extracted and analyzed from root samples. The dried stem, leaf, and fruit were comminuted to dried powders separately in a pharmaceutical disintegrator and sieved through a 100-mesh sieve. The essential oils of dried stem, leaf, and fruit were extracted with petroleum ether at a designed time, temperature, power, and solid–liquid ratio through the ultrasonic-assisted extraction method. The supernatant was collected by centrifugal extraction solution (1200 rpm for 3 min). Essential oils were analyzed on a Thermo GC-MS Trace U3000 system using a chromatographic column, TG-5MS (30 m × 0.25 mm × 0.25 $\mu$m). The oven column heating procedure had been slightly modified to maintain the initial temperature of 50 °C for two minutes and then increased to 180 °C at a rate of 6 °C/min (*Wang et al., 2018*). The mass spectra were obtained by automatic scanning at m/z 35–550 amu after injection of 1.0 $\mu$L sample. Components were identified based on matching their recorded mass spectra to the main library database, and their relative concentrations were calculated by comparing their GC peak areas to the total area.

The extraction of fruit essential oil was carried out according to the optimal extraction method (the solid–liquid ratio was 1:10, 30 °C, 240 W ultrasonic for 30 min) obtained from the orthogonal optimization experiment conducted in the laboratory earlier (*Wang et al., 2018*). The orthogonal $L_9(3)^4$ design was used to extract essential oils from the stem and leaf, and the optimum extraction process was studied. Four factors with three variation levels were listed in Table S2. The experimental conditions and extraction rates for each test and the results of the analysis of variance were presented in Tables S3 and S4. The yield of essential oil (%) was calculated by dividing the essential oil content by the weight of the dried pretreated sample weight. The contents of essential oils extracted from the stem, leaf, and fruit by the optimum extracting technology reached 4.87%, 5.42%, and 6.33%, respectively, which were the average values of the triplicate experiments.

## Statistical analysis

Principal component analysis (PCoA) and hierarchical cluster analysis were performed to evaluate differences of rhizospheric soil and four parts (root, stem, leaf, and fruit) in species complexity. The obtained OTU classification information was used to plot the structure and composition histogram of each sample and the visual heatmap. Finally, the metabolic and ecologically relevant functions of endophyte and rhizospheric soil microorganisms were annotated by Tax4Fun for 16S rDNA OTUs and FunGuild for ITS OTUs. Spearman correlation analysis was mainly used to reflect the relationship between the active secondary metabolites of the sample and the changes in the relative abundance of microorganisms (top 20) in different plant organs. The *p*-values were corrected by the BY method to adjust the error detection rate. All experiments were independently replicated at least three times, and all data were expressed as mean ± standard error. One-way analysis of variance (ANOVA) was used to analyze the differences among secondary metabolites with a significance of $p < 0.05$. All statistical analyses were carried out using R 4.3.1 (*R Core Team, 2023*).

**Table 1  Alpha diversity index of bacteria and fungi (m ± sd).**

| | Alpha index | Rhizosphere soil | Root | Stem | Leaf | Fruit |
|---|---|---|---|---|---|---|
| **Bacteria** | ACE | 74802.77 ± 931.55 a | 7336.57 ± 679.33 b | 8624.06 ± 787.37 b | 5099.82 ± 824.41 c | 5380.41 ± 817.88 c |
| | Chao1 | 70527.98 ± 875.83 a | 8624.06 ± 745.22 b | 4257.31 ± 562.71 c | 4964.72 ± 765.88 c | 4413.24 ± 357.68 c |
| | Shannon | 11.59 ± 0.23 a | 5.97 ± 1.32 b | 4.97 ± 2.13 b | 4.68 ± 0.78 b | 4.48 ± 0.64 b |
| | Simpson | 1.00 ± 0.00 a | 0.81 ± 0.15 a | 0.84 ± 0.17 a | 0.80 ± 0.16 a | 0.78 ± 0.22 a |
| **Fungi** | ACE | 1502.15 ± 312.88 bc | 1184.42 ± 307.64 c | 1989.66 ± 268.35 b | 2541.61 ± 323.54 a | 1210.33 ± 237.67 c |
| | Chao1 | 1555.06 ± 244.68 c | 1192.01 ± 277.37 c | 2142.18 ± 245.33 b | 2659.47 ± 356.57 a | 1283.38 ± 231.64 c |
| | Shannon | 5.15 ± 0.40 bc | 4.85 ± 0.61 c | 6.21 ± 0.51 b | 7.39 ± 0.78 a | 5.69 ± 0.56 bc |
| | Simpson | 0.93 ± 0.05 ab | 0.90 ± 0.04 b | 0.96 ± 0.03 a | 0.98 ± 0.01 a | 0.93 ± 0.02 ab |

**Notes.**
Different lowercase letters (a–c) indicate significant differences ($p$ <0.05) of the same alpha diversity index between rhizosphere soil and different parts of *S. sphenanthera*, one-way ANOVA, Tukey test.

## RESULTS

### OTU analysis and the diversity of bacteria and fungi

To obtain the microbial taxa distribution among plant organs (root, stem, leaf, and fruit), and rhizospheric soil, we analyzed the microbiomes by 16S/ITS amplicon sequencing analysis. The total OTUs of bacteria were significantly different from that of fungi, especially in rhizospheric soil (total OTUs of bacteria were 17 times more than fungi) (Fig. S1A). OTUs cluster analysis showed that the microorganisms in the above-ground parts (stem, leaf, and fruit) of *S. sphenanthera* were grouped into one group, and the microorganisms in the below-ground parts (rhizospheric soil and root) into another category (Figs. S1B and S1C). The Venn diagram results showed that there were differences in composition among rhizospheric soil and different parts of *S. sphenanthera* (Figs. S1D and S1E). There were 10 common OTUs of bacteria in rhizospheric soil and four parts (root, stem, leaf, and fruit) of *S. sphenanthera* (Fig. S1D), which belonged to three phyla (Proteobacteria, Actinobacteria, and Cyanobacteria) respectively. Among the fungi of *S. sphenanthera*, there were 6 common OTUs in rhizospheric soil and four parts (Fig. S1E) belonging to two phyla (Ascomycota and unclassified_Fungi).

Alpha diversity indices were applied for the estimating of endophytic community complexity based on the OTUs sequence, and the diversity of the community was further reflected (Table 1). Among the endophytic and rhizosphere bacteria identified, the diversity of the below-ground parts was higher than the above-ground parts, and the soil Shannon index was nearly two times as high as that of the endophytes index. It indicated that the abundance of microbial species in the rhizospheric soil was high and distributed evenly. Among the identified fungal microbial communities, the Simpson index was above 0.9, the Shannon indexes in the stem and leaf were higher, and the microbial diversity of the fungi was lower than that of the bacteria (Table 1). Chao1 also revealed that the diversity of root samples was higher than that in the leaf samples. On the contrary, fungi (Chao 1) showed an increasing trend of species richness from root to stem and leaf samples (Table 1). The diversity of endophytic microbial in different parts of the host and the specificity of the rhizospheric soil were fully reflected.

The Beta diversity of the microbial community structure of rhizospheric soil and four parts of *S. sphenanthera* was evaluated according to the principal coordinate analysis diagram, which could intuitively show the microbial community difference of rhizospheric soil and different parts (Fig. S2). The distribution of the three points representing stem, leaf, and fruit in the figure was not close to each other, and the separation distance was not very large, which illustrated that the endophyte community structures in the above-ground parts were different, but the differences were small. The part of above-ground parts (stem, leaf, and fruit) were far apart from the below-ground parts (root and rhizospheric soil), indicating that there were great differences in the composition of microbial between the above-ground parts and the below-ground parts (Fig. S2).

## The microbial composition and community structure of bacteria and fungi

The histogram abundance map could analyze the composition and proportion of species between rhizospheric soil and different parts more intuitively, taking the level of the phylum as an example (Fig. 1). The endophytic bacteria were mainly composed of Cyanobacteria and Proteobacteria, and the bacteria of rhizospheric soil were mainly composed of Acidobacteria and Proteobacteria (Fig. 1A). The fungi in rhizospheric soil and different parts of *S. sphenanthera* were mainly composed of Ascomycota and Basidiomycota (Fig. 1B). Ascomycota had higher abundance in fruit, stem, leaf, and root, while Basidiomycota had higher abundance in rhizospheric soil (Fig. 1B). However, there were differences in the dominant flora of endophytes at lower levels in the root, stem, leaf, and fruit (Fig. 2, Figs. S3 and S4, Tables S5 and S6).

There were 12 common bacterial genera in the rhizospheric soil and four parts of *S. sphenanthera* (Fig. 2A). *Achromobacter* was the dominant genus of leaf and stem, and *Methylobacterium* was the dominant genus of leaf. In addition, *Methylobacterium* was also found in fruit and stem (Fig. 2A). *Rhodoplanes* was the dominant genus in root and rhizospheric soil, and only existed in below-ground parts (Table S5). There were 11 common fungal genera belonging to rhizospheric soil and four parts of *S. sphenanthera* (Fig. 2B). Among the common genera, *Alternaria* was found with high abundance in root, stem, leaf, and fruit, especially in the stem. *Cercospora* was found with high abundance in stem and leaf (Fig. 2B). In addition, *Hebeloma* (rhizospheric soil), *unclassified_Helotiales* (root), *Stomiopeltis* (stem), and *Botrytis* (fruit) ran to greater than 10% in their respective domains (Table S6).

The heat map of species level clustering showed that rhizospheric soil and root were still clustered as one group, indicating that microbes of rhizospheric soil and root were relatively close (Fig. S4). The clustering results of rhizospheric soil and four parts were consistent with OTUs clustering results (Fig. S1C). It indicated that rhizospheric soil and root microbes were relatively close, and the microbes of stem, leaf, and fruit were close (Fig. S4). According to the above results, there were significant differences in the composition and diversity of microflora in rhizospheric soil and different parts.

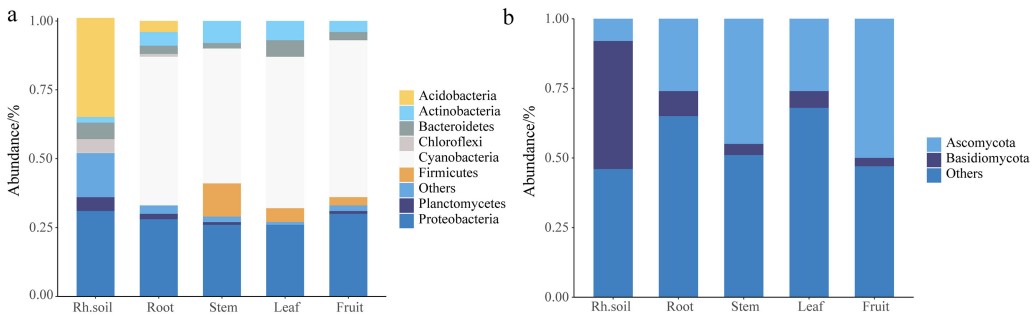

**Figure 1 Stack diagram at the level of the phylum of the sample.** (A) Bacteria. (B) Fungi. Rh.soil, Rhizosphere soil.

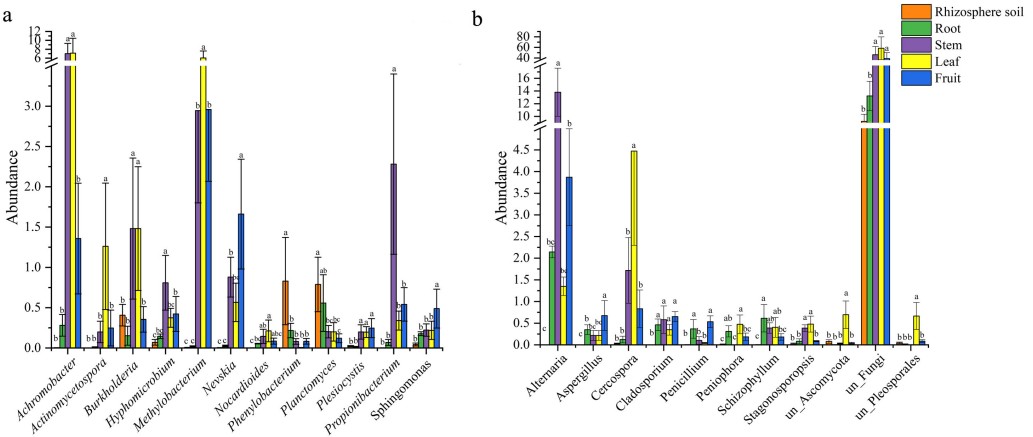

**Figure 2 The differences in the abundance of the common genus of microbial in rhizosphere soil and different parts of _S. sphenanthera_ (m ± sd).** (A) Bacteria. (B) Fungal. Different lowercase letters (a–c) indicate significant differences ($p < 0.05$) of the same genera between rhizosphere soil and different parts of _S. sphenanthera_, one-way ANOVA, Tukey test.

## Functional annotations

To explore the function of microorganisms in different tissue parts of _S. sphenanthera_, Tax4Fun and FunGuild software were used to predict the functions of bacteria and fungi based on 16S rDNA and ITS sequences (Fig. 3). The results showed that there was no significant difference in the function of bacteria between rhizospheric soil and different parts of _S. sphenanthera_, and most of the bacteria were concentrated in metabolism, environmental information processing, cellular processes and so on (Fig. 3A). Among them, more than 70 percent of the bacteria were involved in metabolic function, and among metabolic functions, global and overview maps accounted for about half (Fig. 3A, and Fig. S5). Compared with other parts, the relative abundance of bacteria about two functions (xenobiotics biodegradation and metabolism (Fig. S5A), and microbial metabolism in diverse environments (Fig. S5B)) in rhizospheric soil and root was higher. In addition, the relative abundance of bacteria in root about lipid metabolism, metabolism

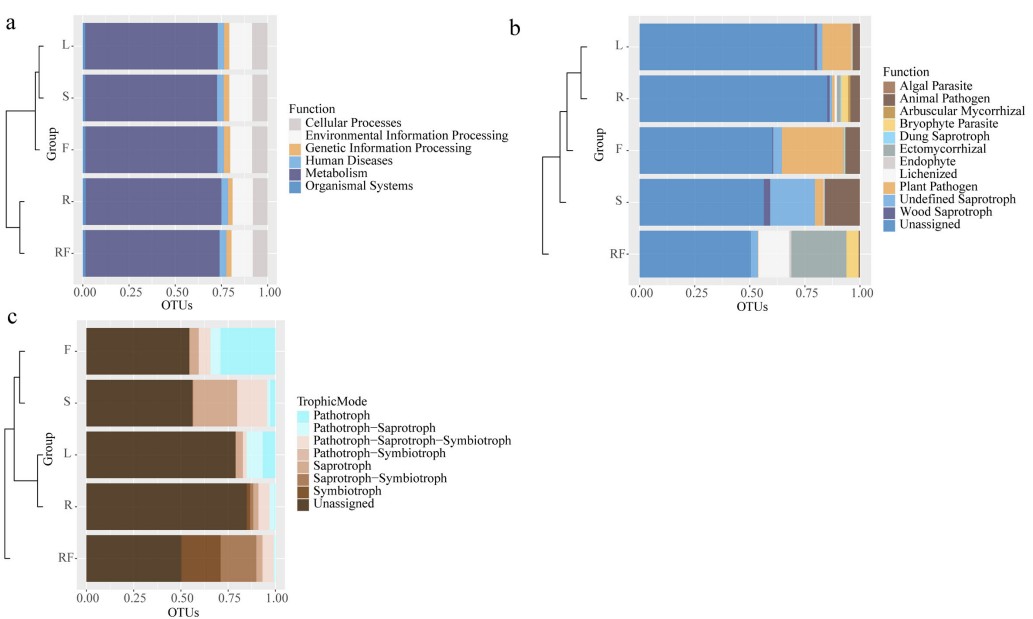

**Figure 3** Horizontal clustering tree-stacking diagram of microbial function annotation (A: bacteria, B: fungal) and trophic mode of fungal (C) in rhizosphere soil and different parts of *S. sphenanthera*. RF, Rhizosphere soil; R, Root; S, Stem; L, Leaf; and F, Fruit.

of terpenoids and polyketides, and fatty acid metabolism was higher than that in other parts (Fig. S5).

However, there were differences in fungal functions between rhizospheric soil and different parts of *S. sphenanthera* (Fig. 3B). Besides the unidentified groups, the fungi in rhizospheric soil and different parts of *S. sphenanthera* included 11 ecological functional groups (Fig. 3B). The relative abundance of fungi about four functions (arbuscular mycorrhizal, bryophyte parasite, ectomycorrhizal, and lichenized) in below-ground parts were much higher than that in the other three parts. Among the four functions, only the relative abundance of fungi with arbuscular mycorrhizal function was greater in root than in rhizospheric soil. Endophytic fungi (root, stem, leaf, and fruit) were mainly pathogenic functional groups and saprophytes, among which the ecological functional groups with higher relative abundance were mainly plant pathogen, animal pathogen, wood saprotroph, and undefined saprotroph. Only endophytic fungi in stem had algal parasite function (0.25%) (Fig. 3B). The results also showed that the fungal groups in the samples were mainly predicted to have seven nutrient types, such as pathotroph, symbiotroph, saprotroph and pathotroph-saprotroph (Fig. 3C). In the rhizospheric soil, saprotroph-symbiotroph and symbiotroph were the main nutrient forms, accounting for more than 39%. Pathotroph was the main nutrient form in the leaf (6.70%) and fruit (29.08%), while the relative abundance of the fungal pathotroph-saprotroph trophic form in the leaf was also high (8.52%). Pathotroph-saprotroph-symbiotroph and saprotroph were the main trophic forms of stem endophytic fungi, accounting for more than 38%. And
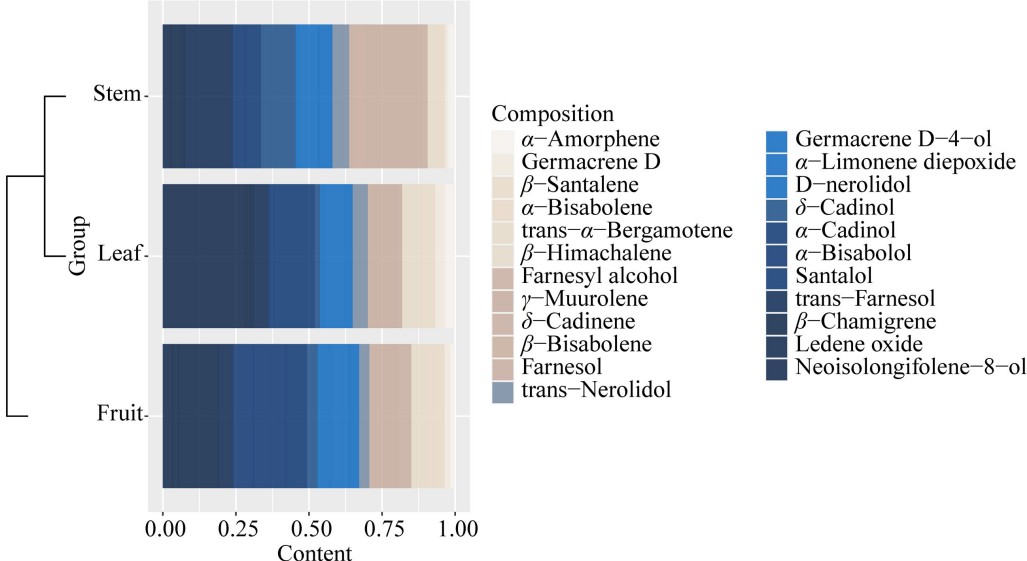

**Figure 4** Common chemical compositions (sesquiterpenes and oxygenated sesquiterpenes) identified in the volatile oils of different parts of *S. sphenanthera*.

the highest trophic pattern of endophytic fungi in the root was pathotroph-saprotroph-symbiotroph (5.96%) (Fig. 3C).

## Content of active secondary metabolites in *S. sphenanthera*

The optimum extraction method for the highest extraction rate of the stem (1:20 ratio, 30 min, 30 °C and 300 W) and leaf (1:15 ratio, 20 min, 35 °C, 240 W) were distinct (Tables S2 and S3). Except that ultrasonic temperature had a small impact on the leaf, other factors had a significant influence on the yield of essential oils from the stem and leaf (Table S4). The factors affecting the yield of essential oils from stem and leaf were listed in order as follows: ultrasonic time >ratio of material to solvent >ultrasonic temperature >ultrasonic power, and the ratio of material to solvent >ultrasonic time >ultrasonic power >ultrasonic temperature (Table S3).

It could be found that the components and content of essential oil varied greatly from stem, leaf, and fruit (Fig. S6, Table S7). The components of essential oils from all samples with low content (<0.1%) were not listed in Table S7. A total of 40 components were identified from all the samples, and 23 essential oil components were identified from the three parts (stem, leaf, and fruit) of *S. sphenanthera*, but the contents were different (Fig. 4 and Table S7). Each part had its high content of dominant components, but there were obvious differences in the dominant components of the three parts. For example, $\gamma$-muurolene, $\delta$-cadinol, and trans-farnesol were characteristic components of the stem. $\alpha$-Cadinol and neoisolongifolene-8-ol were characteristic components of leaf (Fig. 4). Isospathulenol, $\alpha$-santalol, cedrenol, and longiverbenone were characteristic components of fruit (Table S7).

## Correlation analysis

The correlation analysis between microbial communities (top 20 at genus level) and the common components (23) in different parts (Figs. S7 and S8) showed that 12 genera of bacteria and fungi had a significant correlation with the common components ($p < 0.05$) (Fig. 5). Of the 23 essential oil components, only 16 components were associated with bacteria and 10 components were related to fungi (Fig. 5). *Staphylococcus* and *Hyphomicrobium* had positive correlation with five common components ($\gamma$-muurolene, $\beta$-bisabolene, germacrene D-4-ol, trans-farnesol, and ledene oxide). In addition, *Staphylococcus* showed a positive correlation with $\delta$-cadinene ($p < 0.05$), and *Hyphomicrobium* showed a significant positive correlation with $\beta$-bisabolene and ledene oxide ($p < 0.01$) (Fig. 5A). *Methylobacterium* and *Propionibacterium* were correlated with the contents of three components, respectively. *Achromobacter*, *Burkholderia*, and *Planctomyces* had positive effects on farnesol and negative effects on $\beta$-chamigrene, and there was a significant positive correlation among the three genera ($p < 0.01$). Except for the bacteria genera mentioned above, the remaining genera were only related to one component (Fig. 5A). Among the genera of fungi, only *Stomiopeltis* had the most influence on the essential oil components (farnesyl alcohol, $\delta$-cadinene, and $\delta$-cadinol). *Botrytis*, *Cortinarius*, and *Zygophiala* had negative effects on $\alpha$-amorphene and positive effects on $\beta$-chamigrene, and there was a significant positive correlation relationship among the three genera ($p < 0.01$). *Unclassified_Agaricales*, *unclassified_Thelephoraceae*, and *unclassified_Chaetothyriales* had a negative correlation with $\alpha$-bisabolene, and there was a positive correlation relationship between the three genera. In addition, except germacrene D, $\beta$-himachalene, $\alpha$-cadinol, and $\alpha$-bisabolol were affected by only one genus, and the remaining components were associated with two or three genera (Fig. 5B). These results suggested that bacterial communities played a greater role than fungal communities in the accumulation of active secondary metabolites.

## DISCUSSION

The study of endophytes of medicinal plants provided the possibility of screening high-quality strains and fermentation for the production of drug-active ingredients and established a new mode of genuine identification of medicinal materials, which had gradually become a research focus of microbial resources of medicinal plants (*Adeleke & Babalola, 2021*). In this study, Proteobacteria, Cyanobacteria, and Acidobacteria were the main bacteria, and Ascomycota and Basidiomycota were the main fungi at the phylum level (Fig. 1). According to research found that most microbial communities had little difference at the phylum level, which was consistent with the results of this study, which fully demonstrated the similarity of microbes in larger taxonomic units (*Sun et al., 2022*; *Shao et al., 2023*; *You et al., 2021*). At present, endophytes can be isolated from various parts and organs of studied plants, and the structure composition and abundance of endophytes would change with different plant varieties, parts, and development periods (*Araújo et al., 2020*). For example, *Liang et al. (2021)* reported that the fungal diversity of *Codonopsis pilosula* was higher in the leaf part than in the roots and stems, whereas the

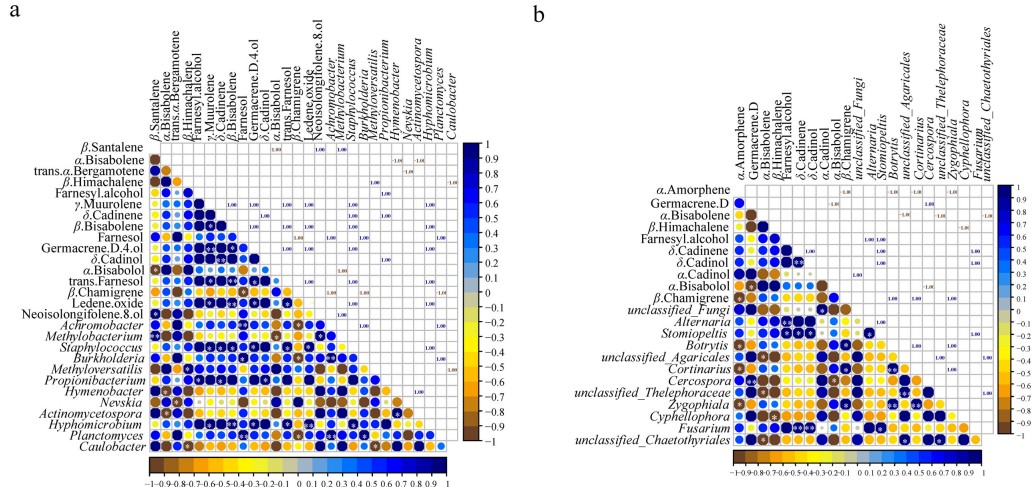

**Figure 5** **Pearson correlations between genera of microbial and common chemical compositions of *S. sphenanthera*.** Asterisks (*, **, ***) indicate significant correlation estimates at the level of 0.05, 0.01, and 0.001, respectively.

fungal communities' richness was higher in the roots. However, we found that the fungal communities' richness of leaves was higher than that of roots and stems in *S. sphenanthera* (Table 1). We speculated that the selection of fungal endophytes was influenced by the plant species (*Mathew et al., 2023*). As for endophytic bacteria, the bacterial diversity and bacterial communities' richness of root was higher than that of stems and leaves in *S. sphenanthera* (Table 1), which was similar to the results of *Liang et al. (2021)*.

Recent studies have reported that root-inhabiting endophytes can selectively come from the soil, and the endophytes of leaves can be selectively derived from the air (*Jia et al., 2018*). Of course, because of the limited entry points on the surface of the leaves, most of the endophytes in the leaves come from the soil (*Jia et al., 2018*). This study revealed that there were significant differences in the community structure and composition of endophytes in different parts of *S. sphenanthera* (Figs. S1B and S1C), which might be related to the different physiological structures of different tissues and the diversity of endophyte sources. The microbial communities of rhizospheric soil and root were more similar than those of endophytes from stem, leaf, and fruit, suggesting that endophytes of root invaded from rhizospheric soil and reached the root through transport tissue. However, the endophytic communities of the stem and fruit were similar to those of leaves, suggesting that the microorganisms of the stem and fruit were influenced by the entry of microorganisms into the phyllosphere in addition to root transport. As the most direct contact part of plants, rhizosphere soil is an active field for material and energy exchange, and its microbial community (rhizosphere microbial community) is a key factor affecting the relationship between plants, soil, and microorganisms (*Li et al., 2018*). Therefore, the diversity and abundance of bacterial communities in rhizosphere soil were the highest, but that of fungi was low, which was consistent with *Hu et al. (2023)*.

As a traditional Chinese medicine, the dried fruit of *S. sphenanthera* was the main part to be used as medicine. Compared with stem and leaf, there were many studies on the essential oil of fruit. The main components of essential oil were sesquiterpenoids and oxidized sesquiterpenoids, which were the same as those reported in previous literature (*Yu et al., 2017*). However, the non-medicinal parts (stem and leaf) were abundant and usually had the same or similar biological active ingredients as the medicinal parts, so they had broad development potential and application prospects. The dominant components of different parts were different (Table S7), and the biological activities of essential oils were related to their main components. The contents of $\gamma$-muurolene, $\delta$-cadinol, and trans-farnesol were the highest in the essential oil of stem. $\gamma$-Muurolene, $\delta$-cadinol, and farnesol were the main components of *Magnolia champaca, Syzygium caryophyllatum*, and *Madhuca Longifolia*, respectively, and all of them have anti-microbial activity (*Henriques et al., 2007*; *Raj et al., 2016*; *Sahoo et al., 2022*; *Silva et al., 2009*; *Suryawanshi & Mokat, 2019*). Moreover, the dominant components in the leaves ($\alpha$-cadinol and neoisonolene-8-ol) also had antibacterial activity (*Hassan et al., 2020*; *Hsu, Su & Ho, 2020*). Among the four dominant components of fruit, isospathulenol as the main component of *Salvia syriaca* essential oil, exhibited strong cytotoxicity, antioxidant, a-amylase, and a-glucosidase inhibitory activities (*Bahadori et al., 2017*). In addition, $\alpha$-santalol had antibacterial activity (*Bommareddy et al., 2018*), cedrenol could be used as a flavorful ingredient (*Bhatia et al., 2008*), and longiverbenone was an active toxic compound (*Khani & Heydarian, 2014*).

Plant secondary metabolites are important agents of plant-microbial interaction (*Sasse, Martinoia & Northen, 2018*). Numerous studies have reported that endophytes can produce various bioactive secondary metabolites that are identical or similar to host secondary metabolites (*Ludwig-Müller, 2015*). It was important to explore the relationship and influence of microorganisms on the quality of *S. sphenanthera*. In this study, we found that the content of secondary metabolites in *S. sphenanthera* was correlated with the presence of microorganisms. Bacteria had a stronger correlation with the accumulation of active components than fungi (Fig. 5). These results highlighted the involvement of specific bacterial communities in plant secondary metabolic pathways, suggesting that a variety of bacterial flora may promote the production of plant secondary metabolites (*Wu et al., 2021*).

Among the secondary metabolites, $\beta$-chamigrene was significantly positively correlated with three endophytic bacteria and negatively correlated with three endophytic fungi (Fig. 5). The results indicated that the synthesis of plant secondary metabolites is related to a variety of endophytes, not just one endophyte (*Hou et al., 2022*). However, the secondary metabolites of *Rheum palmatum* were only correlated with endophytic fungi (*Chen et al., 2021*). The reason for this difference may be related to different plant species. Moreover, *Achromobacter* and *Methylobacterium*, which were correlated with essential oil components (Fig. 5A), had a high abundance in the above-ground part of *S. sphenanthera* (Fig. 2A). The growth environment of wild *S. sphenanthera* is bad, which poses some challenges to its growth. These two bacteria were not only dominant bacteria, but also beneficial microorganisms. *Achromobacter* can use aromatic compounds as the only carbon source, and *Methylobacterium* protects plants from pathogens (*Christian et al., 2021*; *Ho, Hsieh &*

*Huang, 2013*). In addition, *Rhodoplanes, Candidatus Solibacter,* and *Gemmata* were also the dominant flora in the below-ground parts (Table S5). *Rhodoplanes* were facultative photoorganics and potential nitrate fixation bacteria (*Hiraishi & Ueda, 1994*). *Candidatus Solibacter, Gemmata,* and *Staphylococcus* were pathogenic (*Backman & Sikora, 2008*; *Muriuki, Rengan & Budambula, 2021*; *Othman et al., 2021*). However, pathogens could be used as inducers to enhance the host's resistance to disease, activate the host's defense system, and improve the host's defense ability against pathogens (*Backman & Sikora, 2008*). The remaining bacteria were involved in plant protection, growth promotion, and functional secondary metabolites. The results also confirmed that endophytic bacteria and rhizospheric soil microorganisms had a low risk to plants.

The relative abundances of *unclassified_Thelephoraceae, unclassified_Agaricales,* and *Cortinarius* were higher in the below-ground part of the 12 fungal genera that influenced the components of essential oil, while the relative abundances of other genera were higher in the above-ground part (Fig. 5B, Table S6). Some of the fungi in this study were pathogenic microorganisms, and although they temporarily lost pathogenicity, they might regain it under environmental selection, such as *Botrytis* (*Kan, Shaw & Grant-downton, 2014*) and *Zygophiala* (*Batzer et al., 2008*). In addition, some endophytic fungi and their metabolites in above-ground parts could promote the growth and development of host plants, such as *Trametes* and *Eremothecium* (Table S6). Although *Trametes* was a wood saprotroph and *Eremothecium* was a plant pathogen (Fig. 4B), the secondary metabolites of *Eremothecium* and *Trametes* had medicinal value (*Semenova et al., 2022*; *Zmitrovich, Ezhov & Wasser, 2012*). Moreover, *Hebeloma* and *Clavulinopsis* were dominant genera in the rhizospheric soil of *S. sphenanthera* (Table S6). Studies have shown that *Hebeloma* and *Clavulinopsis* are functional fungi involved in soil material transformation, they can transform organic matter and promote soil nitrogen fixation (*Birkebak et al., 2013*; *Mayor, Schuur & Henkel, 2009*; *Mrnka et al., 2020*). These microorganisms were associated with the accumulation of secondary metabolites in *S. sphenanthera*. This may also be one of the reasons for the accumulation of secondary metabolites in wild *S. sphenanthera*.

Because of the roles of endophytes in plant growth, our research results helped expand the application of endophytes in the production of *S. sphenanthera* and its important metabolites. The information on the differences in endophytes between above-ground and below-ground parts could serve as the basis for the selection of functional microorganisms.

## CONCLUSIONS

In conclusion, this is the first study to analyze the microbial diversity in different parts and rhizospheric soil of *S. sphenanthera* and the relationship between microorganisms and secondary metabolites. Our findings demonstrated that the components of essential oils in different parts of *S. sphenanthera* were different, especially the dominant components. At the same time, plant organs have important effects on the composition and structure of the rhizosphere and endophytic communities of *S. sphenanthera*. In addition, positive and negative correlations were found among active secondary metabolites and microbial operating taxonomic units based on the correlation coefficient matrix, indicating that

the synthesis of secondary metabolites in *S. sphenanthera* was closely related to microbial community composition. As a traditional Chinese medicine, *S. sphenanthera* had important practical significance in medicine and the economy. Our findings provide new insights into the distribution and resources of endophytic communities in different parts of the medicine plant. There are still some limitations in this study; the migration mechanism between microorganisms in different parts and how microorganisms affect the active secondary metabolites are not very clear, and further experiments are needed to solve these problems.

### Funding

This work was supported by the Fundamental Research Funds for Central Universities (No. GK202304022), the Natural Science Basic Research Program of Shaanxi Province (No. 2020JM-277), and the National Natural Science Foundation of China (No. 31070293). The funders had no role in study design, data collection and analysis, decision to publish, or preparation of the manuscript.

### Grant Disclosures

The following grant information was disclosed by the authors:
Fundamental Research Funds for Central Universities: GK202304022.
Natural Science Basic Research Program of Shaanxi Province: 2020JM-277.
National Natural Science Foundation of China: 31070293.

### Competing Interests

The authors declare there are no competing interests.

### Author Contributions

- Xiaolu Qin conceived and designed the experiments, performed the experiments, analyzed the data, prepared figures and/or tables, authored or reviewed drafts of the article, and approved the final draft.
- Han Pu performed the experiments, prepared figures and/or tables, and approved the final draft.
- Xilin Fang performed the experiments, prepared figures and/or tables, and approved the final draft.
- Qianqian Shang analyzed the data, prepared figures and/or tables, and approved the final draft.
- Jianhua Li analyzed the data, prepared figures and/or tables, and approved the final draft.
- Qiaozhu Zhao performed the experiments, prepared figures and/or tables, and approved the final draft.
- Xiaorui Wang conceived and designed the experiments, performed the experiments, analyzed the data, authored or reviewed drafts of the article, and approved the final draft.

- Wei Gu conceived and designed the experiments, authored or reviewed drafts of the article, and approved the final draft.

## Data Availability

Sequences are available at https://figshare.com/articles/dataset/totalwild_fa/22214602.

## Supplemental Information

Supplemental information for this article can be found online at http://dx.doi.org/10.7717/peerj.17240#supplemental-information.

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
