# Peer review of "Microbial communities of Schisandra sphenanthera Rehd. et Wils. and the correlations between microbial community and the active secondary metabolites"

_PeerJ, doi:10.7717/peerj.17240_

## Round 0.1 · original submission · Major Revisions

Several concerns regarding statistical analysis have been raised by one of the reviewers. Besides all the comments of reviewers, the authors should focus on the statistical analysis and language of the manuscript.

**Language Note:** The review process has identified that the English language must be improved. PeerJ can provide language editing services - please contact us at [email protected] for pricing (be sure to provide your manuscript number and title). Alternatively, you should make your own arrangements to improve the language quality and provide details in your response letter. – PeerJ Staff

Reviewer 1 ·

Basic reporting

Language contains minor errors, which are not disturbing and do not interfere with interpretation.

Intro OK.

Structure OK.

Figures: see my detailed suggestions for authors.

Raw data. One of the provided links (https://www.ncbi.nlm.nih.gov/bioproject/938991) does not contain raw data.

Experimental design

Scope OK.

Well defined research question, in my opinion, microbe - metabolite correlations are of interest.

There are experimental design and data interpretation issues.
See my major points in additional comments.

There are data presentation issues.
See my major points in additional comments.

Validity of the findings

The conclusions of the current version of the paper are supported by the research to a limited extent only.
See my major points on experimental design and data processing for details.

Additional comments

Dear Authors,

Please find a list of suggestions to be addressed. The "major points" are sorted in decreasing order of importance, but they are all to be dealt with.

Best regards.

Major points.

1., False discovery rate. Statistical inference was carried out on large datasets with a p < 0.05 for each feature which is not acceptable as a large number of false positives are mistaken as being significant. Therefore, a false discovery rate adjustment is required for p-values. This is especially important for correlation analyses and will render several currently p < 0.05 phenomena insignificant. As a consequence, sections L298-320 and the respective discussion will need an update. I suggest using the BY correction (?p.adjust in R).

2., There is no mention of using a negative control in high-throughput sequencing. Check (Hornung et al., 2019). There is no mention of solvent blanks, process blanks, sample storage, randomization, etc. in the GC-MS protocol.

3., There was no "metabolism quenching" (Álvarez-Sánchez et al., 2010), samples were air dried. Note this might acceptable for plant materials with essential oil constituents, depending on what the ambient conditions are (if so, add a reference!), but this is unlikely. This is for sure unacceptable for general metabolomics, especially primary metabolism and possibly also for thick organs. Liquid N2 treatment followed by lyophilization is preferred.

4., It is a pity that root essential oil was not examined. A higher number of biological replicates (plants) should have been used, which would have enabled studying this without harming the wild population of plants. This raises the following question: What was actually fed into correlation analyses? All data points from all organs, pooled? n = ?

5., L308 (and possibly elsewhere): they positively correlated, correlation can be an indicator of, but in itself is not a causal relationship. If the essential oil (EO) can come in contact with microbes during asymptomatic colonization (this might depend on the anatomical structures that contain the EO in Schisandra), negatives correlations are easily interpreted as these compounds tend to be antimicrobial agents. For a set of possible interpretations of correlations, check the discussion of Plaszkó et al. (2022).

6., Metabolite annotation. The reported limonene diepoxide would be an extremely unusual compound for and EO - make sure you report only compounds with high conviction database matches. I think a 750-800 threshold is usually appropriate (on a scale of 1000). Your drying method (leaving the plant to dry) might have influenced the amount of oxidized terpenoids (Abbas et al., 2021). Additionally, limonene diepoxide is a monoterpene, so your claim in L291 might not hold. Also note that it is possible to feed unknown but nicely integrated peak AUC values into correlations.

7., Are the prerequisites of ANOVAs met (normality)? Perhaps Kruskal-Wallis tests are to be used.

8., L250-260: If there were no significant differences, there should be no discussion about differences, as they are to be considered noise.

9., Fig. 3c is misleading - why is there no "unassigned" group? The "unassigned" category should have been inherited from a previous level of annotation, I guess.

Minor issues.

10., Rhizosphere definition. I think 2 mm diameter roots unlikely hold large fragments of attached soil, but this depends on the soil sand/silt/clay content. In order to be sure that your sampling complies with the definition of the rhizosphere (e.g. Tkacz et al. (2020)), add method details on soil layer thickness, and/or soil particle size distribution data.

11., M&M Section "Statistical analysis" contains mostly the bioinfromatics pipeline, which is not statistical analysis. The "Statistical analysis" section is important, it should be about as replicate numbers, statistical methods and software, p-value adjustment for false discovery rate / family-wise error rate.

12., Orthogonal array (OA) extraction design. The "optimal" set of conditions are always present in the original OA matrix. Is this a case by luck or no modeling was carried out to obtain an optimal combination?
L163-165: numbers do not match those for optimal parameter combination in Tables S3-4. Why?

13., Supplementary and other figure captions and issues.
13a., Fig. 3: An abbreviation (R, S, L, F, RS) might be more appropriate than an A-E coding.
13b., Fig. S3: Despite "others" is present, not all stacked barplots add up to 1.00. This is most likely because averages were attempted to be summed, which obviously do not exactly add up to 1.00. Consider calculation of the proportion of "others" for each sample separately, summing the proportions and finally divide by the number of samples. Some subplots have % (0-100) and other have proportion (0-1). Fix this as well. Additionaly, after fixing, this plot will be much more interesting than Fig.1., consider replacing them.
13c., Fig. S4: I guess this is some autoscaled dataset, but the caption does not tell this. Add more details on what is plotted.
13d., Fig. S7-S8: Data should be ordered along hierarchical clustering. This will mix the bacterial OTUs and the EO components, but will reveal clusters with multi-correlation. FDR has to be used here as well (in fact, it is most important here). The diagonal should be empty, it has no biological/chemical meaning.
13e., Table S4: Add experiment to title. Table of variance analysis of ... .
13f., Table S6: footnote unclear. What criteria were used to include 69 of more than 100 OTUs?
13g., Table S7: What is the unit of measure for the compounds? If these are % of all peak areas, these are compositional data and have to be pretreated before using them in correlation studies, as they contain negative correlations by definition .
13h., L190: These Venn diagrams do not show genetic relationships, only composition

14., L233 and possibly elsewhere: microorganisms do not have a "concentration", but "abundance" or "proportion"

15., L342: Please add more relevant articles on similarity / dissimilarity of microbes, possibly ones done in close relatives of Schisandra.

16., L357-359: you are studying petrolether extracts and not EOs. Color comes from extracted non-volatiles like chlorophylls. The identified EO compounds are colorless.

17., L368-378: most of human applications have no relevance to the study. Antimicrobial effects are.

18., Section "Effects of microorganisms on active secondary metabolites of S. sphenanthera": for several sentences, it is unclear what plant system the authors are talking about. I suggest expanding towards microbes that accept sesquiterpenes as carbon sources, and data on variability of antimicrobial acivity of sesquiterpenes. Citations linking to medicinal values or industrial uses of the pure strains is less relevant in my opinion and they might be removed from the paper.

References

Abbas, A.M., Seddik, M.A., Gahory, A.-A., Salaheldin, S., Soliman, W.S., 2021. Differences in the Aroma Profile of Chamomile (Matricaria chamomilla L.) after Different Drying Conditions. Sustainability 13, 5083. https://doi.org/10.3390/su13095083
Álvarez-Sánchez, B., Priego-Capote, F., Castro, M.D.L. de, 2010. Metabolomics analysis II. Preparation of biological samples prior to detection. TrAC Trends in Analytical Chemistry 29, 120–127. https://doi.org/10.1016/j.trac.2009.12.004
Hornung, B.V.H., Zwittink, R.D., Kuijper, E.J., 2019. Issues and current standards of controls in microbiome research. FEMS Microbiology Ecology 95, fiz045. https://doi.org/10.1093/femsec/fiz045
Plaszkó, T., Szűcs, Z., Cziáky, Z., Ács-Szabó, L., Csoma, H., Géczi, L., Vasas, G., Gonda, S., 2022. Correlations Between the Metabolome and the Endophytic Fungal Metagenome Suggests Importance of Various Metabolite Classes in Community Assembly in Horseradish (Armoracia rusticana, Brassicaceae) Roots. Frontiers in Plant Science 13.
Tkacz, A., Bestion, E., Bo, Z., Hortala, M., Poole, P.S., 2020. Influence of plant fraction, soil, and plant species on microbiota: A multikingdom comparison. mBio 11. https://doi.org/10.1128/mBio.02785-19

·

Basic reporting

Language could be more concise and clear. Notes are provided on the PDF.
Abstract needs help. Including specific results, even though they will be few, will make the abstract more interesting and your article more visible.
A few notes made in the introduction.
Methods are ambiguous with regards to essential oils.
Results are relevant.
Discussion is confusing and conclusion is strong.

Experimental design

Needs to be more clear about the number of replications.
Methods are ambiguous for essential oil section.

Validity of the findings

The statistical soundness was not emphasized and could be highlighted more.

Additional comments

Please consider highlighting a few significant results in the abstract, discussion, and conclusion. Other results do not need to be removed from the manuscript but we need to know what the big picture is.
Writing this out concisely is one of the challenges of connecting these related studies in a cohesive manuscript that will be impactful and widely read.

Reviewer 3 ·

Basic reporting

no comment

Experimental design

1) The sampling protocol for the rhizospheric soil seems not scientific, and some important information are missing. The standardized method could refer to Barillot et al (2013, Annals of microbiology).

Validity of the findings

2) The characterization of the soil microbiota was performed, but the data of root chemicals was missing.
3) The functional analysis should be used to generate hypotheses, not to draw any conclusions given the limitations of such an indirect approach. This method heavily relies on proper gene annotation and could also have significant pitfalls when considering strain-level functional differences that are not resolved based on 16S profiling.
4) In order to assess whether the microbiota contributed to the accumulation of plant secondary metabolites, I suggest to either remove or exchange the microbiota in plant or soil. As this might be beyond the scope of this manuscript, I suggest that the authors at least acknowledge the limitations of the descriptive and in vitro approaches, and also word more carefully their conclusions. The currently reported data are insufficient to draw conclusions on compound synthesis symbiosis from a microbiologist’s point of view.

Additional comments

Qin et al. investigate the relationship between the active secondary metabolites and the microorganisms of S. sphenanthera. They perform phylogenetic analysis and diversity of rhizospheric soil microorganisms and endophytes in different parts of S. sphenanthera. Also they measure the secondary chemicals in different parts. The claim to demonstrate the correlation between the content of with plant chemicals with its microorganisms that potentially contributed to the accumulation of plant secondary metabolites. The research question of the authors is interesting and timely. In recent years, there is increasing evidence that plant-associated microbes play a role in chemical biosynthesis in plant and some interesting results have been obtained by researchers in this field.

The authors of this manuscript question whether the rhizospheric soil and plant microbiome of S. sphenanthera mediates accumulation of plant secondary metabolites. They have provided extensive data analyses of the obtained 16S and ITS amplicon data giving more insight into the compositions of the microbiomes of rhizospheric soil and plant.

Unfortunately, in my opinion the authors were not able to answer their research question since they have not shown that the microbiomes positively and negatively affect the accumulation of plant secondary metabolites directly though they analyze the correlation between microbial communities (Top 20 at genus level) and the common components in different parts.

Annotated reviews are not available for download in order to protect the identity of reviewers who chose to remain anonymous.

---

## Round 0.2 · Minor Revisions

I would recommend acceptance of this article. However, several minor comments of reviewer 2 need to be addressed before formal acceptance of the current article.

Reviewer 1 ·

Basic reporting

no comment

Experimental design

Apart from my comment #1, the current version is acceptable.

Validity of the findings

Apart from my comment #4, the current version is acceptable.

Additional comments

Though I see overall improvement when rereading the paper, in my opinion, some of my (and other referees') questions have not been adequately answered. Therefore, I'll use the numbers from my previous opinion.

The provided line numbers do not match that of the final PDF (what do they match?), so I sought changes in the document with "track changes" enabled.

Major issues.

1., The statistical correction method and its scope must appear in M&M section. Also, the term "adjusted p value" should be added to Figure captions where appropriate.

4., When I asked what was actually fed into correlation analyses, I wanted to know how many points are used to establish a correlation between a chemical and a microbial feature (n = ?). Did you use all data points from all organs, pooled? As the chemical differences between organs as well as the microbial community sources are obvious, organs are unlikely good candidates for data pooling.

·

Basic reporting

Several sentences were rewritten which has improved the clarity of the manuscript. This article could still benefit from additional proofreading or English language editing. Several sentences require a second reading.

The abstract still does not contain any main results that would capture the reader's attention:
Example:
The contents of essential oil components and lignans in fruit were much higher than that in stem and leaf, and the dominant essential oil components in different parts were different.

What was the main component for each of the 3? The sentence lacks content.

Example:
Results showed that the accumulation of secondary metabolites in S. sphenanthera was closely related to the microbial community composition, especially bacteria.

It would be more informative to note one or 2 of the main taxa that was related to each of the 3 tissues studied. Or if you want to keep it simple, give the taxa with the most compelling association to the terpene with the highest cor value. Or the results which seem to have the most biological significance can be given. The most interesting result from your paper is in lines 374-382. If you can please summarize that section in 1-2 sentences and add that to the abstract and conclusion, that will be sufficient.

Experimental design

No comment.

Validity of the findings

For the unclassified reads, you should justify why you are keeping them in. For example, does keeping in the unclassified reads assist with quantifying the fungi your abundance analysis? Do you think that these reads could be from novel organisms? It would be interesting to see how many different representative sequences make up these unclassified reads.

Additional comments

The efforts of the authors to improve the draft are evident. A few changes are required to improve the clarity and visibility of the manuscript.

Reviewer 3 ·

Basic reporting

The author have improved the MS according to the previous comments carefully. No comment any more.

Experimental design

No comment

Validity of the findings

No comment

Additional comments

No comment

---

## Round 0.3 · accepted · Accept

All queries have been addressed in the revised version.